# Experimental Investigation on the Relationship Between COD Degradation and Hydrodynamic Conditions in Urban Rivers

**DOI:** 10.3390/ijerph16183447

**Published:** 2019-09-17

**Authors:** Lei Tang, Xiangdong Pan, Jingjie Feng, Xunchi Pu, Ruifeng Liang, Ran Li, Kefeng Li

**Affiliations:** State Key Laboratory of Hydraulics and Mountain River Engineering, Sichuan University, Chengdu 610065, China; tanglei961@163.com (L.T.); PanXiangdong0506@163.com (X.P.); puxunchi@scu.edu.cn (X.P.); liangruifeng@scu.edu.cn (R.L.); liran@scu.edu.cn (R.L.); kefengli@scu.edu.cn (K.L.)

**Keywords:** urban rivers, COD, degradation coefficient, hydraulic conditions, dimensional analysis

## Abstract

Due to extensive pollution and the relatively weak flow replacement in urban rivers, determining how to fully utilize the self-purification abilities of water bodies for water quality protection has been a complex and popular topic of research and social concern. Organic pollution is an important type of urban river pollution, and COD (chemical oxygen demand) is one of the key pollution factors. Currently, there is a lack of research on the relationship between COD degradation and the flow characteristics of urban rivers. In this paper, COD degradation experiments were conducted in an annular flume with Jinjiang River water at controlled flow velocities and the COD degradation coefficients under different hydraulic conditions were analyzed. A good correlation was observed between the degradation coefficient and hydraulic conditions. According to dimensional analysis, the relationship between the COD degradation coefficient and hydraulic conditions such as the flow velocity, water depth, Reynolds number (*Re*), and Froude number (*Fr*) was established as $K_{\mathrm{COD}}=86400\frac{u}{h}Fr^{0.8415}Re^{-1.2719}+0.258$. The COD degradation coefficients of the Chishui River in Guizhou Province ranged from 0.175–0.373 1/d based on this formula, and the field-measured values varied from 0.234–0.463 1/d. The error in the formula ranged from 5.4–25.3%. This study provides a scientific basis for the prediction of the COD degradation coefficients of urban rivers.

## 1. Introduction

Many urban rivers exhibit high channelization degrees, shallow water depths, small longitudinal gradient, and weak water exchange effects [1]. Pollution of urban rivers has become increasingly problematic in recent years [2], and organic pollution is an important type of urban river pollution [3]. Research on organic pollution involves COD (chemical oxygen demand) degradation [4,5], which refers to a decrease in pollutants in the water column through physical, chemical and biological processes [6]. The degradation coefficient is an important parameter that represents the rate of degradation process, and is affected by many factors, such as the pollutant components [7], hydraulic conditions [8], water temperature [9], pH [10], microbial properties [11], dissolved oxygen [12] and sediment properties [13]. Among these factors, hydraulic conditions strongly influence the COD degradation coefficient of urban rivers. 

Common methods for determining the COD degradation coefficient include empirical formula estimations, data analyses, indoor experiments and field observation methods [14,15]. Empirical methods are often developed from data collected at a limited number of rivers and river conditions, which makes them difficult to generalize. In turn, indoor experimental methods have high requirements for water testing, flow conditions and pollutant scenarios. Thus, the experimental conditions may be different from those in the natural environment, which means that errors may arise. Field observation methods are based on the measured water quality data obtained in the field. However, the pollution source of a river may be difficult to identify, large observed datasets are required, and work can be difficult to perform in some river sections. Generally, empirical methods and data analysis methods have low precision, which may lead to unreliable results. Although indoor experimental methods have high accuracy, urban rivers are susceptible to the effects of different pollutant sources, which can be difficult to represent in laboratory conditions. In view of the limitations of the empirical, data analysis and field methods, indoor experimental methods are often used to determine the pollution degradation coefficient in practical applications. 

To study pollutant degradation coefficients and degradation rate estimates based on indoor experimental methods, some researchers used simulated wastewater [16] without considering the pollutant characteristics of the natural river. Such conditions result in large differences between the natural pollutant characteristics and flow conditions of urban rivers and those of the experiment, which caused large prediction errors and uncertainty. Wang et al. (2007) [17] simulated the process of TN(total nitrogen) degradation in a rectangular channel using untreated freshwater from the Qinhuai River and discussed the relationships between the TN self-purification coefficient and the flow velocity, Reynolds number, and Froude number. The experimental results showed that smaller the Reynolds numbers and Froude numbers corresponded to a greater the degradation rate of TN, but a quantitative relationship between the degradation coefficient and hydraulic characteristics was not established. Huang et al. (2017) [14] simulated the degradation of COD and NH_3_-N at different flow velocities in a straight circulating flume by sampling the raw water from the Beijing River, which is influenced by non-point source pollution and domestic pollution. The results showed that the relationship between the COD degradation coefficient and flow velocity was linear and that between NH_3_-N and flow velocity was a power function. The fitting formula is as follows:(1)$$K_{\mathrm{COD}}=0.0726u+0.0089$$
(2)$$K_{{\mathrm{NH}}_{3}\text{-}N}=0.2293u^{0.3036}+0.038$$
where $K_{\mathrm{COD}}$ (1/d) represents the COD degradation coefficient, $K_{{\mathrm{NH}}_{3}\text{-}N}$ (1/d) represents the NH_3_-N degradation coefficient, and *u* (m/s) represents the velocity. The result considers the influence of the flow velocity on the degradation coefficient but neglects the effects of other hydraulic conditions, such as water depth, Reynolds number, and Froude number. With respect to an infinite flow velocity, the degradation coefficient calculated by the formula will increase to an indefinite value, indicating that the formula can only be applied in a certain range of flow velocities, a limitation that is not discussed in the study. 

According to previous studies, the relationship between the COD degradation coefficient and hydraulic conditions of natural rivers is not appropriate to solve practical problems. In this study, we sampled raw water from the Jinjiang River to conduct experiments that simulate different hydraulic conditions in an indoor annular flume with an adjustable flow velocity. The relationship between the COD degradation process and hydraulic characteristics under different hydraulic conditions was investigated. 

## 2. Materials and Methods 

### 2.1. Experimental Device

Experiments were conducted at the State Key Laboratory of Hydraulics and Mountain River Engineering (SKLH) at the Sichuan University of China. To maintain constant temperature conditions, experiments were conducted in a constant-temperature laboratory. An annular flume that was 0.2 m wide and 0.3 m high was used to simulate natural river conditions. Flow at the annular flume was driven by an LDZ100-125D centrifugal pump. The outlet of the pump was equipped with a perforated plate to ensure smooth flow conditions. The experimental device is shown in Figure 1.

### 2.2. Experimental Methods and Materials

The experimental water was sampled from the Jinjiang River, and the sampling site was located in Guojiaqiao, Wuhou District, Chengdu, as shown in Figure 2. Natural water was used in the experiment, and the characteristics of the pollutants and microbial species in the original river were considered to reflect the degradation trend of organic matter under natural conditions. The sampled Jinjiang River water was drained into the annular flume and the static tank, and the temperature was controlled at 20 °C in the constant-temperature laboratory.

### 2.3. Methods for Monitoring and Measurement

The water temperature was monitored by a ZDR-21 temperature recorder produced by Zhejiang Top Instrument Equipment Co., Ltd. of China. When the flow temperature stabilized, the centrifugal pump was adjusted to obtain the required flow velocity. The flow velocity was monitored by a NKY02-1C current meter produced by Nanjing Hydraulic Research Institute of China. The experiment was started after the flow velocity stabilized. Water samples were taken from the sampling cross-section in each annular flume at different times. Measurements of COD and DO (dissolved oxygen) were performed using the chemical oxygen demand-dichromate method (HJ 828-2017) and dissolved oxygen-iodometric method (GB 7489-87), respectively. To ensure the accuracy of the data, each sample was tested twice. The relative standard deviation was below 5%, and the data in this paper were the average values of the two tests.

The collected datum of the upstream and downstream sections of the urban river were analyzed by prototype observation method, and the COD degradation coefficient of the river reach can be calculated by the following formula [18]:(3)$$K=\frac{\ln L_{0}/L}{t}$$

The formula can also be expressed as follows:(4)$$K=\frac{86400\left(\ln L_{0}-\ln L\right)u}{x}$$
where *x* (m) represents the length of the calculated reach, *u* (m/s) represents the flow velocity, *K* (1/d) represents the pollutant degradation coefficient, *L*_0_ (mg/L) represents the pollutant concentration of the upstream section, and *L* (mg/L) represents the pollutant concentration of the downstream section.

### 2.4. Setup of Scenarios

The COD degradation processes were composed by physical, chemical and biological processes. Considering the effect of biochemical reaction, we set the scenarios of tiny flow velocity. Furthermore, we set different flow conditions to study the role of hydraulic conditions to the COD degradation coefficient. 

To make the results more representative, Scenarios 1–5 were conducted on the same date, and Scenarios 6–10 were conducted on different dates. The conditions of the experimental scenarios are shown in Table 1.

## 3. Results

### 3.1. The COD Degradation Process and Degradation Coefficient

The variation in the COD concentration with reaction time during degradation is shown in Figure 3. The COD concentration gradually decreases with time, and the degradation rate exhibits a trend from fast to slow. The figure shows that the degradation rate of COD is slower at a low flow rate. The final degradation levels of COD at different flow velocities are 62.5–66.0% (u = 0.001 m/s), 77.6–76.9% (u = 0.10 m/s), 79.8–81.2% (u = 0.15 m/s), 77.5–81.8%. (u = 0.20 m/s), and 78.2–85.7% (u = 0.30 m/s).

It is generally accepted that the degradation process of organic matter conforms to a first-order kinetic equation [19,20], and the form of this equation can be expressed as follows:(5)$$\frac{dC}{dt}=-KC$$

By integrating both sides of the formula, the equation can be rewritten as follows:(6)$$C=C_{0}e^{-Kt}$$
where *t* (d) represents the reaction time, *K* (1/d) represents the pollutant degradation coefficient, *C* (mg/L) represents the pollutant concentration at time *t*, and *C*_0_ (mg/L) represents the initial concentration of pollutants.

Considering the influence of temperature on the pollutant degradation coefficient, experiments were conducted in a constant-temperature laboratory. In the laboratory, the temperature distribution was not uniform and temperature fluctuations occurred during the experiments. Scenario 1–Scenario 5 were conducted during the same period, and differences in temperature occurred in these scenarios. Scenario 6–Scenario 10 were conducted in different periods, and the temperature was controlled stably at 20 °C in each scenario.

According to the monitored temperature for each scenario, the fitted degradation coefficients were corrected by the Phelps empirical formula [21]:(7)$$K_{T}=K_{20}\theta^{\left(T-20\right)}$$
where *K*_T_ (1/d) represents the pollutant degradation coefficient at temperature T, *K*_20_ (1/d) represents the pollutant degradation coefficient at 20 °C, and *θ* (1.047) represents the temperature correction coefficient.

The COD degradation process followed a first-order kinetic equation, and the fitted degradation coefficient was corrected according to the temperature correction coefficient. The obtained COD degradation coefficients are shown in Table 2. The COD degradation coefficients increase with increasing flow velocity, and the relationship between the COD degradation coefficient and the flow velocity is shown in Figure 4. As the flow velocity was 0.001 m/s, the COD degradation coefficients varied from 0.228 1/d–0.271 1/d. When the flow velocity increased to 0.30 m/s, the COD degradation coefficients ranged from 0.394 1/d–0.397 1/d. As the flow rate continued to increase, the effect of the flow rate on the COD decay rate diminished, which suggests a decrease in the microbiological response. Due to experimental flow rate limitations, no experiments with high flow conditions were performed.

### 3.2. Analysis of the Relationship Between the Degradation Coefficient and Hydraulic Conditions

Considering the hydraulic parameters that influence the degradation coefficient, including the fluid density *ρ* (ML^−3^), gravitational acceleration *g* (LT^−2^), water depth *h* (L), flow velocity *u* (LT^−1^), and dynamic viscosity *μ* (ML^−1^T^−1^), the degradation coefficient can be expressed as follows.

(8)K=f(ρ,h,u,g,μ)

A dimensional analysis is performed with *ρ*, *g*, and *h* as fundamental physical quantities, and Equation (8) can be rewritten as follows:(9)$$K\frac{h}{u}=f\left(\frac{gh}{u^{2}},\frac{\mu }{\rho uh}\right)$$

Because the kinematic viscosity *v* (L^2^T^−1^) can be expressed as $\frac{\mu }{\rho }$, and the left side can be kept only *K*, the equation can be again rewritten as follows:(10)$$K=\frac{u}{h}f\left(\frac{gh}{u^{2}},\frac{\nu }{uh}\right)$$
where $\frac{gh}{u^{2}}$ can be expressed as a function of the Froude number (*Fr*) and $\frac{\nu }{uh}$ can be expressed as a function of the Reynolds number (*Re*). Thus, the function of *K* can be expressed as follows:(11)$$K=\frac{u}{h}f\left(Fr,Re\right)$$

The unit of $\frac{u}{h}$ is 1/s, and the right side must be multiplied by 86,400 to transfer the unit of *K* to 1/d. The dimensional analysis yields the following equation:(12)$$K=86,400\frac{u}{h}Fr^{a}Re^{b}+c$$
where $\frac{u}{h}$ represents the ratio of the flow velocity to the water depth (T^−1^), *Fr* represents the ratio of the inertia force to the gravitational force, and *Re* represents the ratio of the inertia force to the viscous force.

To obtain the relationship between the degradation coefficient and hydraulic conditions, 10 scenarios were investigated. The experimental parameters are shown in Table 3. A regression analysis was employed to analyze the data in Table 3. The optimal values of a, b and c were 0.8415, −1.2719 and 0.258, respectively, and the correlation coefficient of the regression equation was 0.78 (see Figure 5). By substituting the values into Equation (12), the formula for the COD degradation coefficient can be expressed as follows:(13)$$K_{\mathrm{COD}}=86400\frac{u}{h}Fr^{0.8415}Re^{-1.2719}+0.258$$

### 3.3. Verification of the Formula Using Field Observation Data

To verify the accuracy of the relationship between the obtained degradation coefficient and hydraulic conditions, the Chishui River in Renhuai City, Guizhou Province is selected for observation. This urban river has conditions similar to the sampled river, the river reach is located in a natural protection area of valuable and rare fish of the Yangtze River, and the shape of the river section is regular. Additionally, the water flow conditions are relatively mild, and there are few pollution sources. The COD degradation coefficient of the river section was analyzed using the monitoring data collected at the Renhuai Environmental Protection Monitoring Station on 3 January 2017, 6 April 2017, and 3 July 2017, which represent the wet season, temperate season and dry season, respectively. The hydraulic conditions and water quality observation results are shown in Table 4.

The COD degradation coefficients of the Chishui River reach in different periods were obtained by substituting the parameters in Table 4 into Equation (1), and the calculated degradation coefficients were corrected for temperature by Equation (7). The corrected degradation coefficients and the calculated degradation coefficients based on field observations are shown in Table 5.

The error between the COD degradation coefficient of the river reach in the wet season, temperate season and dry season calculated by the formula and obtained by the field observations were 25.3%, 5.4%, and 19.5%, respectively. The COD degradation coefficients calculated by the formula were smaller than the values obtained by field observation.

## 4. Discussion

In this study, we analyzed the relationship between the COD degradation coefficient and hydraulic conditions. Due to the complexity of natural rivers, there are many environmental factors related to hydraulic conditions that can affect the COD degradation coefficients, which leads to difficulty and errors in determining the COD degradation coefficient.

### 4.1. Influence of the DO Concentration

DO is one of the most important variables that affects water quality. Low concentrations of DO may directly affect the water quality and disrupt a healthy ecological balance [22]. Hydraulic conditions have a significant impact on dispersion and reaeration within a river [23]. When the flow velocity is 0.001 m/s, the exchange of oxygen between air and water is weak, resulting in a poor reaeration effect in the water body. For slightly faster flowing water, the exchange of oxygen between water and air is accelerated due to the role of turbulence, which leads to a high level DO concentration in water. In the hydrodynamic scenarios (other than u = 0.001 m/s), there was no obvious difference in the DO concentration, and all values remained at approximately 8 mg/L. The variation of the DO concentration in different scenarios is shown in Figure 6.

### 4.2. The Effect of Biochemical Process 

Although the hydraulic conditions are important factors affecting the degradation coefficient, the reduction of the amount of COD concentration in the annular flume is a direct reflection of the biochemical reaction. On the one hand, the river water provides a suitable environment for the metabolism of microbes [24]. On the other hand, the increase of the flow velocity enhances the frequency of contact between microorganisms and pollutants, and accelerates the biodegradation process of pollutants [14]. At a tiny flow velocity (u = 0.001 m/s), the COD degradation coefficients are 0.228 1/d–0.271 1/d, which can almost reflect the biochemical process of the COD degradation. As the flow velocity rises to 0.30 m/s, the COD degradation coefficients are increased by 1.45–1.74 times compared with that at a tiny flow velocity. It shows that hydraulic conditions promote the process of COD degradation, which is related to biochemical effects. When applied to other rivers, the promotion of biochemical effects by hydraulic conditions in different rivers should be considered.

### 4.3. Different Source of Water 

COD was primarily removed via microorganism biodegradation [25]. Different sources of pollution can lead to different microbial community compositions, which may result in large differences in the COD degradation coefficient [26]. The COD degradation coefficient obtained under the experimental conditions ranged from 0.228 1/d–0.397 1/d depending on the temperature. For comparison, Huang et al. (2017) [14] calculated COD degradation coefficients ranging from 0.011 1/d–0.071 1/d in a straight indoor flume. This range is obviously smaller than that obtained in this paper. The raw water from the Beijing River used by Huang et al. (2017) was affected by non-point source pollution and domestic pollution, while the sampled water in this paper was only influenced by domestic pollution. The analysis suggested that the river reach is located in the protected area which is less affected by pollutants. The exchange of nutrients in the river can promote the metabolism of microorganisms [27], resulting in rapid COD degradation in rivers.

The concentrations of pollutants also influence the degradation coefficient. When they are collected, they increase pollution levels or introduce new pollutant components that affect the concentration of dissolved oxygen in the river and the composition of the original components of organic matter, which is susceptible to sampling time. Rainfall may occur upstream before samples settle in a more complex pollutant degradation process [28]. Han et al. (1998) [29] established an empirical model for estimating the COD degradation coefficient by analyzing the relationship between river flow velocities and COD concentration coefficients in Shandong Province. The model assumes that the larger the COD concentration is, the larger the COD degradation coefficient becomes. However, the pollution of the sampling area in this research is mainly from domestic pollution, which consists of some fixed pollutants, and the initial concentration of the pollutant does not differ significantly over time; therefore, the impact of the initial COD concentration on the degradation coefficient is not considered. In practical applications, different pollution occurs, such as urban non-point source pollution, irrigation backwater pollution, livestock breeding pollution and garbage pollution [30]. Moreover, the influence of the initial concentrations of pollutants should be comprehensively considered due to regional differences. 

### 4.4. Content of Sediment 

The presence of sediment accelerates the degradation process of organic pollutants [13]. Due to the adsorption of organic matter by sediment, the degradation coefficient of the organic matter in sandiness water will be larger. According to previous studies [31,32], the sediment content of the Jinjiang River and the Chishui River is low; thus, the predicted value is consistent with the observed value. When the empirical equation is applied to a river with a high content of sediment, the predicted COD degradation might be smaller than the observed value, which can lead to simulation errors. 

### 4.5. Uncertainty of the Results

Although natural water from the Jinjiang River was used in the experiment to reflect the degradation trends in natural rivers, the initial concentration range of pollutants in the experiment was limited due to the specific pollution source. Hence, the obtained trend in the paper has a certain scope of application. In this study, the variation in the degradation coefficient under hydrodynamic conditions is discussed. However, the hydrodynamic force of circulated water was derived from a pump in the indoor experiment and the experiments are conduct in the circular flume with a DC length of 4m and an arc length of 1.4 m, which can be different from the curving ratio of the natural river, thus resulting in reduced accuracy of the prediction. 

## 5. Conclusions and Prospects

This study analyzed the impact of different hydraulic conditions on COD degradation coefficients. The effect of increasing the flow velocity on the COD degradation process was mainly reflected in biological changes, and the COD degradation coefficient increased with increasing flow velocity. Using a dimensional analysis, an empirical formula was established for the COD degradation coefficient considering hydraulic parameters, such as flow velocity, water depth, Reynolds number and the Froude number. The empirical formula was verified using field observation data from the Chishui River. The differences between the degradation coefficient calculated by the formula and those determined based on field observations were 24.8%, 3.0% and 19.4%.

In this study, the variation in the COD degradation coefficient under different hydraulic conditions is described, but because of the complex conditions in the natural environment, there are still many other factors that may affect the degradation coefficient, such as the initial concentrations of pollutants, the water temperature and the pH. It is necessary to quantitatively study the effects of more complex factors on COD degradation coefficients of urban rivers in future work. 

## Figures and Tables

**Figure 1 ijerph-16-03447-f001:**
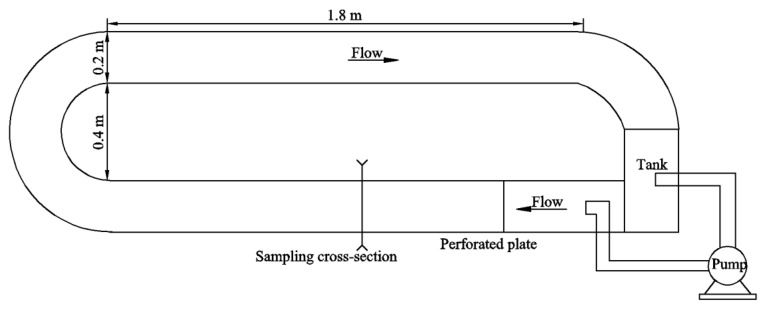
Sketch of the experimental apparatus.

**Figure 2 ijerph-16-03447-f002:**
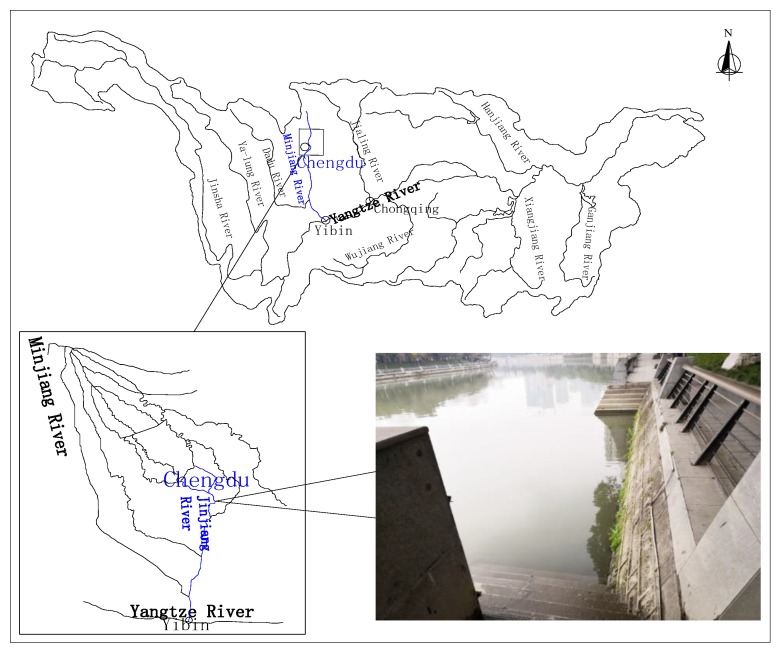
Location of the sampling site.

**Figure 3 ijerph-16-03447-f003:**
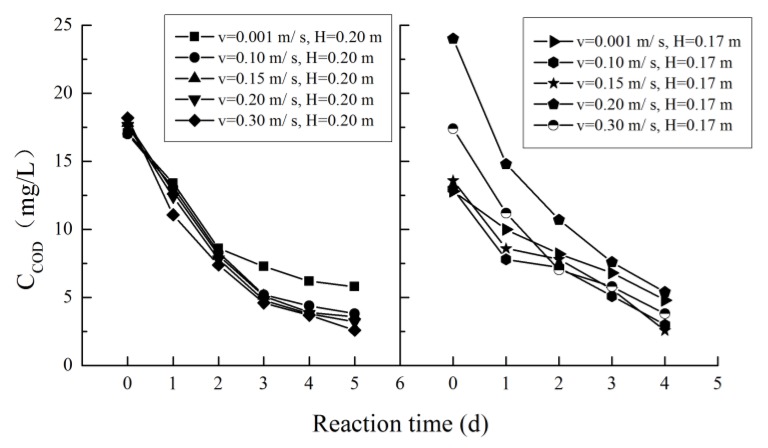
Variation in the chemical oxygen demand (COD) concentration under different hydraulic conditions.

**Figure 4 ijerph-16-03447-f004:**
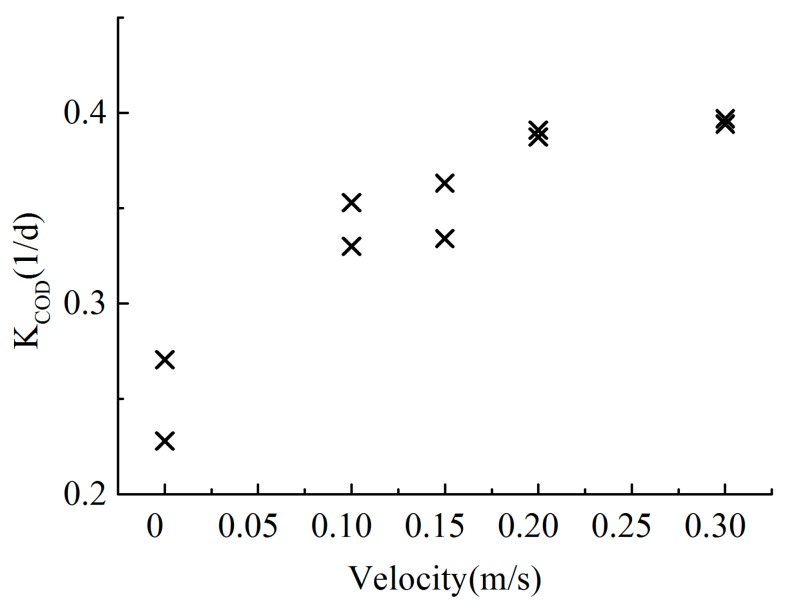
Relationship between *K*_COD_ and the flow velocity.

**Figure 5 ijerph-16-03447-f005:**
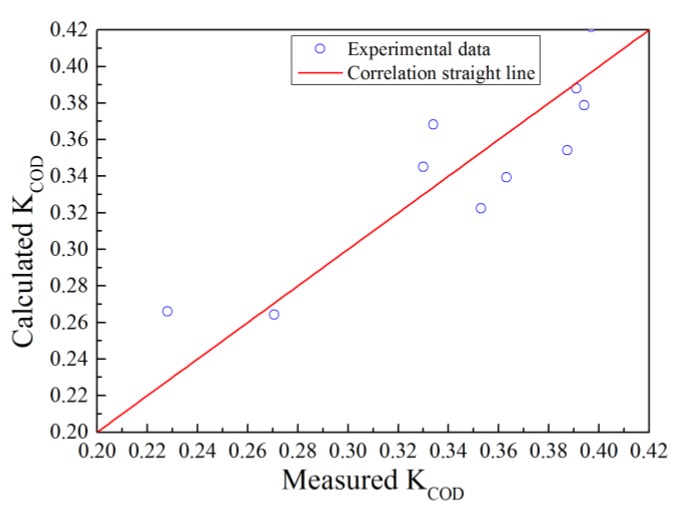
Comparison of the measured and calculated values of the COD degradation coefficient.

**Figure 6 ijerph-16-03447-f006:**
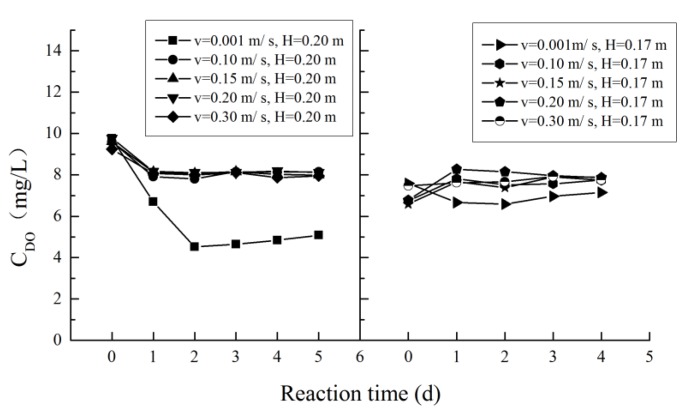
Variation of the dissolved oxygen (DO) concentration.

**Table 1 ijerph-16-03447-t001:** Conditions in the experimental scenarios.

No.	Velocity (m/s)	Water Depth (m)	Sampling Date
Scenario 1	0.001	0.20	23-2-2019
Scenario 2	0.10	0.20	23-2-2019
Scenario 3	0.15	0.20	23-2-2019
Scenario 4	0.20	0.20	23-2-2019
Scenario 5	0.30	0.20	23-2-2019
Scenario 6	0.001	0.17	10-7-2019
Scenario 7	0.10	0.17	16-7-2018
Scenario 8	0.15	0.17	26-7-2018
Scenario 9	0.20	0.17	9-9-2018
Scenario 10	0.30	0.17	1-7-2018

**Table 2 ijerph-16-03447-t002:** Fitting results of *K*_COD_ under different hydrodynamic conditions.

No.	Velocity (m/s)	Initial *C*_COD_ (mg/L)	Water Temperature (°C)	Fitted *K*_COD_ before Temperature Correction (1/d)	Temperature Correction Coefficient	Corrected *K*_COD_ (1/d)
Scenario 1	0.001	17.08	19.0	0.258	1.047	0.271
Scenario 2	0.10	17.00	19.5	0.345	1.023	0.353
Scenario 3	0.15	17.80	20.8	0.377	0.964	0.363
Scenario 4	0.20	17.60	20.2	0.391	0.991	0.387
Scenario 5	0.30	18.20	22.0	0.432	0.912	0.394
Scenario 6	0.001	12.80	20.0	0.228	1.000	0.228
Scenario 7	0.10	12.96	20.0	0.330	1.000	0.330
Scenario 8	0.15	13.60	20.0	0.334	1.000	0.334
Scenario 9	0.20	24.00	20.0	0.391	1.000	0.391
Scenario 10	0.30	17.40	20.0	0.397	1.000	0.397

**Table 3 ijerph-16-03447-t003:** Experimental data.

No.	Initial $C_{\mathrm{COD}}$ (mg/L)	u/h (1/s)	*Fr*	*Re*	$K_{\mathrm{COD}}$ (1/d)
Scenario 1	17.08	0.005	0.001	66	0.271
Scenario 2	17.00	0.500	0.071	6622	0.353
Scenario 3	17.80	0.750	0.107	9933	0.363
Scenario 4	17.60	1.000	0.143	13245	0.387
Scenario 5	18.20	1.500	0.214	19867	0.394
Scenario 6	12.80	0.006	0.001	63	0.228
Scenario 7	12.96	0.588	0.077	6254	0.330
Scenario 8	13.60	0.882	0.116	9382	0.334
Scenario 9	24.00	1.176	0.155	12509	0.391
Scenario 10	17.40	1.765	0.232	18763	0.397

**Table 4 ijerph-16-03447-t004:** Hydraulic conditions and water quality observation results for the Chishui River.

Date	Length of River Reach (km)	Velocity (m/s)	Water Depth (m)	Width (m)	Water Temperature (°C)	*C*_COD_ at the Maotai Section (mg/L)	*C*_COD_ at the Lianghekou Section (mg/L)
3-1-2017	43.4	0.61	0.78	81.6	11.4	17.3	15
6-4-2017	43.4	0.89	1.41	82.8	19.8	16.3	14
3-7-2017	43.4	1.17	2.16	84.3	28.0	18.3	15

**Table 5 ijerph-16-03447-t005:** Comparison between the calculated and observed values.

Date	Calculated *K*_COD_ (1/d)	Measured *K*_COD_ (1/d)	Error
3-1-2017	0.175	0.234	25.3%
6-4-2017	0.256	0.271	5.4%
3-7-2017	0.373	0.463	19.5%

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
