# Peer review of "Experimental Investigation on the Relationship Between COD Degradation and Hydrodynamic Conditions in Urban Rivers"

_ijerph, 2019, doi:10.3390/ijerph16183447_

Round 1
Reviewer 1 Report
The authors did a great job in addressing the issues identified in the paper. I think that the paper has improved significantly and I recommend "Publication after minor revisions". The only issues that I'd like to the authors to address before publication are:
1) Refer to my comment and your response to "Point 10: Line 104-106": Although the emphasis of this research is on the hydraulic effects, I still think that the authors shouldn't ignore completely the fact that thebiogeochemistry can be as important (if not more in some cases) than the hydraulics. In my view, there should be at least 1 to 2 paragraphs in the "Materials and Methods" section discussing the potential effects that these transformations (which ones and how they are mediated) may have in the overall COD dynamics. That information should be used to discuss the limitation of your hydraulic-focus-only approach and could be used to add some comments about future research.
2) Refer to my comment and your response to "Point 21: Eq. 12": I still think that this equation is out of place. It is part of the methodology used and not a finding of this work. Thus, it should not be in "Results" section because here is where the results of this work are presented - its place should be in a subsection under "Material and Method" (in my opinion). The authors just need to add an introductory statement in this sub-section explaining what that equation (developed by others) will be used in this study.
Author Response
Point 1: Refer to my comment and your response to "Point 10: Line 104-106": Although the emphasis of this research is on the hydraulic effects, I still think that the authors shouldn't ignore completely the fact that the biogeochemistry can be as important (if not more in some cases) than the hydraulics. In my view, there should be at least 1 to 2 paragraphs in the "Materials and Methods" section discussing the potential effects that these transformations (which ones and how they are mediated) may have in the overall COD dynamics. That information should be used to discuss the limitation of your hydraulic-focus-only approach and could be used to add some comments about future research.
Response 1: Thanks very much for your suggestion. The degradation process of pollutants is composed of physical, chemical and biological processes. Although the paper mainly discussed the effect of hydraulic conditions in the COD degradation process, we haven’t ignored the importance of biochemical action for degradation. The purpose of the setup of the scenarios was described in the “Materials and Methods” section. And we have supplemented a sub-section to discuss the effects of biochemical process.
Point 2: Refer to my comment and your response to "Point 21: Eq. 12": I still think that this equation is out of place. It is part of the methodology used and not a finding of this work. Thus, it should not be in "Results" section because here is where the results of this work are presented - its place should be in a subsection under "Material and Method" (in my opinion). The authors just need to add an introductory statement in this sub-section explaining what that equation (developed by others) will be used in this study.
Response 2: We feel great thanks for your professional suggestion. And we have adjusted the equation to the “Materials and Methods” section. We appreciate for your work earnestly, and hope that the correction will meet with approval. Once again, thank you very much for your comments and suggestions.

Reviewer 2 Report
The paper concerns COD degradation experiments conducted in an annular flume with Jinjiang River water at controlled flow velocities and the COD degradation coefficients analysis under different hydraulic conditions. Authors have sampled raw water from the Jinjiang River to conduct experiments that simulate different hydraulic conditions in an indoor annular flume with an adjustable flow velocity. The relationship between the COD degradation process and hydraulic characteristics under different hydraulic conditions was investigated. The paper is concise, well written and its scope is very interesting. It is prepared very accurately. The introduction section is appropriate and its introduce the reader to the scientific scope of the manuscript in details. The experimental and results section are also well prepared with a proper discussion and reference to the published so far papers in the similar scientific scope. The weakness of the paper is lack of graphical abstract which should reflect the manuscript content in a more specific way. It would increase the scientific value of the manuscript.
Author Response
Point 1: The paper concerns COD degradation experiments conducted in an annular flume with Jinjiang River water at controlled flow velocities and the COD degradation coefficients analysis under different hydraulic conditions. Authors have sampled raw water from the Jinjiang River to conduct experiments that simulate different hydraulic conditions in an indoor annular flume with an adjustable flow velocity. The relationship between the COD degradation process and hydraulic characteristics under different hydraulic conditions was investigated. The paper is concise, well written and its scope is very interesting. It is prepared very accurately. The introduction section is appropriate and its introduce the reader to the scientific scope of the manuscript in details. The experimental and results section are also well prepared with a proper discussion and reference to the published so far papers in the similar scientific scope. The weakness of the paper is lack of graphical abstract which should reflect the manuscript content in a more specific way. It would increase the scientific value of the manuscript.
Response 1:Thank you very much for your comments on our article. According to your suggestion, we have supplemented the graphical abstract to demonstrate the work of this research . We hope the graphical abstract could provide a guidance to the reader.
This manuscript is a resubmission of an earlier submission. The following is a list of the peer review reports and author responses from that submission.
Round 1
Reviewer 1 Report
Dear Authors,
The paper "Experimental Investigation on the relationship of COD degradation and hydrodynamic conditions of urban rivers" need minor
revision (improved literature, made corrections to minor methodological
errors and text/figure editing).
I think this paper would benefit to be a stronger bridge to other works, so it is necessary to introduce more the link between introduction & results/discussion
So in my opinion the literature need to be improved... is necessary to cite more paper from the last 2-5 years. The recent papers (except few) are quite old of this manuscript.Moreover I strongly recommend to join discussion and perspective rather then conclusion and perspective as well as extend discussion also with more recent papers to compare this study results and others achievement.
Please also make some corrections in Fig. 3 and 4... its something wrong need some point editing, that now are very small and dificult to read!
Author Response
Thanks for your contribution very much about our manuscript, and those comments are very helpful for improving our work. We have revised carefully our manuscript based on the opinions and suggestions.
Point 1: The paper "Experimental Investigation on the relationship of COD degradation and hydrodynamic conditions of urban rivers" need minor revision (improved literature, made corrections to minor methodological errors and text/figure editing).
Response 1: Thank you for your suggestion. We have made as detailed modifications as possible in literature, methodological errors and text/figure editing. Furthermore, in order to improve the logic and language accuracy of the paper, we sent the article to an editing company to make the expression closer to the native speaker.
Point 2: I think this paper would benefit to be a stronger bridge to other works, so it is necessary to introduce more the link between introduction & results/discussion.
Response 2: We have mentioned several methods for determining the degradation coefficients in the introduction. Considering the practicability and applicability, we obtained an empirical formula to determine the degradation coefficient by used indoor experimental method, and the formula was verified by the original observed data. The results show that the method has good feasibility, which can save time and improve efficiency. We hope that the research can provide a better reference for the prediction of river water quality in future. In the introduction and discussion, we also mentioned many factors that could affect the accuracy of the prediction of the degradation coefficient, which is still the problem to be solved in the future.
Point 3: So in my opinion the literature need to be improved... is necessary to cite more paper from the last 2-5 years. The recent papers (except few) are quite old of this manuscript. Moreover I strongly recommend to join discussion and perspective rather than conclusion and perspective as well as extend discussion also with more recent papers to compare this study results and others achievement.
Response 3: Thank you for pointing out the insufficient of the reference of this paper. We have added the latest literature to make the study more timeliness. In the new revised version, we changed the structure of the article to enrich the discussion section.
Point 4: Please also make some corrections in Fig. 3 and 4... its something wrong need some point editing, that now are very small and difficult to read!
Response 4: Thanks for your reminding that the illustration is not clear enough. The content of the picture is not clearly presented. We have revised the Figures to increase the readability.

Reviewer 2 Report
GENERAL COMMENTS
This manuscript investigates the impact of different hydraulic conditions on COD degradation rates in urban rivers. This is accomplished by simulating different hydraulic conditions (more specifically flow velocity and water depth; a total of 10 scenarios) on an annular flume and measuring the temporal dynamics of COD concentrations at a particular cross-section. The experimental results were used to derive an empirical equation via dimensional analysis. The generated formula was subsequently tested on field conditions based on data sampled from the Chishui River. Although the topic is interesting for the readership of IJERPH and the scientific contribution of the manuscript is relevant, I identified the following critical issues: 1) the English is generally poor, particularly in the Discussion and Conclusions section, and 2) the discussion is also very week and full of assertions. The authors need to show more command of the literature in the Discussion section to ensure that the readers understand the exact novel contribution of the paper. Unfortunately, I recommend this paper to be REJECTED because it still needs substantial structural changes before it can be considered for publication. I provide below a detailed analysis of the paper and many suggestions on how to improve the paper. I encourage the authors to re-submit the paper once these issues have been addressed.
GENERAL NOTES
Lines 14 and 33: The statement “organic pollution is the main type of urban river pollution” is unclear. Do you mean the dominant type of pollution? Even if you write “dominant”, the statement is very vague and unlikely generalizable. I suggest changing to “is an important type of in urban…”
Lines 42 to 43: awkward sentence. Suggest changing to something like “Empirical methods are often developed from data collected at a limited number of rivers and river conditions, which makes them difficult to generalize.
Line 55: what is “manually derived wastewater”? It makes no sense; clarify.
Line 72: give examples of “effects of other hydraulic conditions” by adding (for instance) “, such as…” after “conditions”
Line 74: Do the authors in Huang et al. (2017) mention these limitations? If yes, please specify. Otherwise, add something like “, a limitation that is not discussed in the study” after “of flow velocities”
Line 76: What do you mean by “must be sufficiently considered” - vague statement.
Line 97: do you mean “annular flume and static tank? If yes, please correct
Line 104: suggest changing “Cases” by “Scenarios”. If the authors agree with this change, then please replace it throughout the manuscript
Line 104: Also, the authors need to introduce the “representative Cases… “ (or scenarios) before this sentence. I suggest moving the sentence in 105 (“The conditions… Table 1.”) before the sentence in line 104 (“To make…”)
Line 104-106: I find surprising that being the degradation of DOC a bio-mediated process, that the authors don’t discuss anything about microbiological factors and communities – the paper needs more depth in summary and discussion of biogeochemical factors. This makes me wonder how generalizable are the findings in this river to similar (and different) river systems
Line 118-120: The sentences “DO is ones… ecological balance[18]” is not a finding of this work and seems to be out of place. Move it to the Discussion section if appropriate.
Line 122: In the sentence “In flowing water, the exchange of oxygen between water and air is sufficient due to turbulence”; what do you mean by “is sufficient”? Is sufficient for what? Vague statement.
Line 122-123: Where is the information in the sentence “In all the… 8 mg/l” taken from?
Line 132-133: “… the temperature was not uniform…” please discuss the implications that this may have in the results
Line 150: the authors cannot talk about “microbiological response” because this was not measured, but rather inferred from COD decay rates. Suggest changing to “…, on the COD decay rate, which suggests a decrease in the microbiological response”
Table 2: In column 4 for “Stable temperature”, the authors are including all the “cases”. However, it is said in lines 132-133 that the temperature was no uniform. Please clarify.
Figure 4 and Table 1: There are no replicates for each scenario, which gives no guarantee that the experiment is reproducible and the results are consistent and reliable.
Eqs 6 to 11: It is a bit difficult to follow the derivation of the empirical expression. I suggest the authors provide more explanations for each of the steps. For instance, it’s impossible to understand why Eq. 11 is multiplied by 86400 when going from Eq. 10 to Eq. 11
Line 181: “This urban river has conditions similar to the sampled river”. This means that the empirical formula has only been tested on very similar flow, temperature and water quality conditions. How generalizable is the equation to other regions? Is it still valid under different environmental conditions? I understand that the authors cannot possibly contemplate all the possible scenarios in the lab, but this needs to be properly discussed in the “Discussion Section” so that the reader can understand the strengths and weaknesses of the method and formulae proposed.
Line 185: The river was monitored on three different occasions between Jan and June 2017, and this is the data that was used to test the empirical equation on field conditions. Three measurements are clearly insufficient to make any trustworthy assessment of the method.
Eq. 12: This equation is not a finding of this work. Therefore, it should no be in the section “Results”. Instead, it should in a sub-section of “Materials and Methods” that explains the methodology adopted to test the empirical equation in field conditions.
Line 204: The statement “Analysis suggested… degradations in the river” is vague, unclear and not a finding of this work. Move it to the Discussion section if appropriate.
Line 219-223: The authors did not include any analyses on the effect of sedimentation and chemical analysis of the water besides COD. In fact, this is the first time that sedimentation is mentioned in the paper. So, this is not a finding of the paper. It can be used to discuss the results, but the authors need to make a better job at integrating their findings with existing literature on the topic.
Line 223-226: This statement doesn’t make any sense and, in my opinion, is not adequately linked to the results of this paper
Line 230-231: What do you mean by “the pollutant source… is fixed…”? Please be more precise.
Line 238-240: “… is discussed” But this is the section where all should be discussed.
Line 240: what do you mean by “this approach may disrupt the degradation process”? please clearly explain why and how.
TECHNICAL NOTES AND SUGGESTIONS:
Line 17: change “controllable” to “controlled”
Line 31: change “levels” by “depths”
Line 31: unclear “small gradient ratios” means. The ratio of what by what? Clarify
Line 31: what is flow replacement? Clarify.
Line 32: suggest replacing “serious” by “problematic”
Line 39: remove “level”
Line 98: remove “to keep… stable.” – it is unnecessary
Line 39: replace “Amongst these factors” by “In addition to these environmental factors”
Line 40: replace “highly” by “strongly”
Line 42: “estimations” instead of “estimation” AND “analyses” instead of “analysis” AND “experiments” instead of “experiment”
Line 44: suggest adding “In turn, indoor experimental methods…”
Line 45: suggest changing to “… testing, flow conditions and pollution scenarios. Thus, the experimental conditions may be different from…”
Line 51: replace “various” with “different”
Line 51: change to “, which can be difficult to represent in laboratory conditions”
Line 54: change to “The study of pollution degradation and estimation of degradation rates based on…”
Line 58: add “and uncertainty” after “large prediction errors”
Line 60: add “flow” before “velocity”
Line 71: remove “the” before “flow velocity”
Line 71: replace “neglect” by “neglects”
Line 77: suggest replacing “paper” by “study” AND replacing “sample raw” by “sampled raw”
Line 80: change to “was investigated”
Line 84: add “the” before “Sichuan”
Line 87: change to “simulate natural river conditions”
Line 87: Replace “The hydrodynamic force on” by “Flow at” – be direct and concise
Line 89: “add conditions” after “smooth flow”
Line 99: this is a continuation of the previous sentence – remove the paragraph
Line 113: I suggest removing “and analyses of experiment”
Line 132: remove “new paragraph”
Table 4: replace “of” to “at” in the header columns 7 and 8
Table 4: change the date formatting to something more standard and intuitive. For instance, replacing “.” by “-“ would already help. The same things in Table 5
Line 197: There is no Eq. 14. Please correct this.
Line 198: add “for temperature” before “by Eq. (5)”.
Line 2001: Replace “According to the results, it can be seen that the” with simply “The”
Line 210: “many influential factors”? maybe replace with “many environmental factors”?
Line 210: change to “…many environmental factors that can affect the COD degradation coefficients”
Line 212: change to “… 1/d depending on the temperature”
Line 212: change to “calculated COD degradation coefficients ranging…”
Line 217: need to add a reference to the statement “Different sources… COD degradation coefficients”
Line 230: replace “will be” with “becomes”
Line 230: replace “pollutant” with “pollution”
Line 232: replace “much” with “significantly”
Line 232: concentration of what?
Line 234: what pollutants? Vague statement. Please give some examples at least and provide references for the readers to follow up if they desire
Line 242: replace the sentence to something like “This study analyzed the impact of different hydraulic conditions on COD degradation rates/coefficients.
Line 248: remove “of the COD degradation coefficient”. The readers know this by now.
Line 250: add “determined” after “those”
Author Response
We feel great thanks for your professional review work on our article. Those comments are valuable and very helpful for revising and improving our paper, as well as the important guiding significance to our researches. The detailed responses are listed below, and we look forward to your further guidance.
Point 1: Lines 14 and 33: The statement “organic pollution is the main type of urban river pollution” is unclear. Do you mean the dominant type of pollution? Even if you write “dominant”, the statement is very vague and unlikely generalizable. I suggest changing to “is an important type of in urban…”
Response 1: Thanks for your suggestion. We are sorry that the expression is not clear enough and we have revised the statement according to your suggestion.
Point 2: Lines 42 to 43: awkward sentence. Suggest changing to something like “Empirical methods are often developed from data collected at a limited number of rivers and river conditions, which makes them difficult to generalize.”
Response 2: Thanks a lot for your suggestion. We think your sentence is more suitable to describe the insufficient of empirical methods and we have changed the sentence in Line 42-44.
Point 3: Line 55: what is “manually derived wastewater”? It makes no sense; clarify.
Response 3: Thank you for pointing out the inappropriate expression. Some researchers always use self-produced wastewater to simulate different compositions of wastewater to perform specific experiments in laboratory. We have changed “manually derived wastewater” to “simulated wastewater”.
Point 4: Line 72: give examples of “effects of other hydraulic conditions” by adding (for instance) “, such as…” after “conditions”
Response 4: Thank you for your suggestion. We have supplemented the information in Line 73.
Point 5: Line 74: Do the authors in Huang et al. (2017) mention these limitations? If yes, please specify. Otherwise, add something like “, a limitation that is not discussed in the study” after “of flow velocities”
Response 5: Huang et al. (2017) don’t mention these limitations, which are analysed by our observation of the experimental results. We have supplemented the sentence according to your suggestion.
Point 6: Line 76: What do you mean by “must be sufficiently considered” - vague statement.
Response 6: Thanks for pointing out the unclear statement, and we have changed this part in Line 78.
Point 7: Line 97: do you mean “annular flume and static tank? If yes, please correct.
Response 7: Thanks for your reminding and we have revised it.
Point 8: Line 104: suggest changing “Cases” by “Scenarios”. If the authors agree with this change, then please replace it throughout the manuscript.
Response 8: We agree with you. Scenarios are more suitable to describe the experimental conditions than Cases in the study. We have changed it in the manuscript.
Point 9: Also, the authors need to introduce the “representative Cases… “ (or scenarios) before this sentence. I suggest moving the sentence in 105 (“The conditions… Table 1.”) before the sentence in line 104 (“To make…”)
Response 9: Thanks for your suggestion. We have adjusted the order of the sentence.
Point 10: Line 104-106: I find surprising that being the degradation of DOC a bio-mediated process, that the authors don’t discuss anything about microbiological factors and communities – the paper needs more depth in summary and discussion of biogeochemical factors. This makes me wonder how generalizable are the findings in this river to similar (and different) river systems.
Response 10: Your suggestion is very helpful for our work. Actually, the microbiological factors play an important role in the degradation process. However, measuring the microbiologic community is not a part of this study. The study is focusing the influence of hydrodynamic conditions on the COD degradation, so we did not measure the microbiologic community. In the future, we will expand the research to the purification by microorganism.
In the Introduction part, we discuss many urban rivers’ characteristics are high degrees of channelization, shallow water depths, small longitudinal gradient, and weak water exchange effect. Due to the characteristics of urban rivers, the hydraulic conditions are the key factors that influence the COD degradation coefficient. Based on the condition of urban rivers, we mainly consider the characteristics of hydrodynamics. It’s still generalizable to different rivers.
Point 11: Line 118-120: The sentences “DO is ones… ecological balance[18]” is not a finding of this work and seems to be out of place. Move it to the Discussion section if appropriate.
Response 11: Thanks for your suggestion. There is no doubts that DO is an important factor that can influence the COD degradation process, and we have adjusted this part to the Discussion,
Point 12: Line 122: In the sentence “In flowing water, the exchange of oxygen between water and air is sufficient due to turbulence”; what do you mean by “is sufficient”? Is sufficient for what? Vague statement.
Response 12: We are sorry for the improper statement. The turbulence could accelerate the exchange of oxygen between water and air, and that cause a high level DO concentration in water. We have revised in Line 226.
Point 13: Line 122-123: Where is the information in the sentence “In all the… 8 mg/l” taken from?
Response 13: Thanks for your reminding. We measured the variation of the dissolved oxygen in each scenario, but we did not present the data in the paper. We have supplemented variation process of the dissolved oxygen in each scenario in the sub-section of “Discussion”.
Point 14: Line 132-133: “… the temperature was not uniform…” please discuss the implications that this may have in the results
Response 14: Temperature is an important factor influencing the degradation coefficient, and we have taken temperature into account in the study. Due to the temperatures are inconsistent in each scenario, we have implemented the temperature correction formula Eq. (5) when determining the degradation coefficient. The degradation coefficients under each scenario are compared at the same temperature after calibration. So we believe the results are reliable.
Point 15: Line 150: the authors cannot talk about “microbiological response” because this was not measured, but rather inferred from COD decay rates. Suggest changing to “…, on the COD decay rate, which suggests a decrease in the microbiological response”
Response 15: As you suggested, we have changed this part in Line 153-154.
Point 16: Table 2: In column 4 for “Stable temperature”, the authors are including all the “cases”. However, it is said in lines 132-133 that the temperature was no uniform. Please clarify.
Response 16: Due to the structure and system layout of the constant-temperature laboratory, long-term operation will lead to a decrease in temperature control efficiency, resulting non-uniform air distribution in the laboratory. Therefore, the temperature in each scenario is different, and the temperature set by the original experiment is also different. However, the water temperature will remain stable after the exchange of water and air reached equilibrium, and the water temperature recorded is stable.
Point 17: Figure 4 and Table 1: There are no replicates for each scenario, which gives no guarantee that the experiment is reproducible and the results are consistent and reliable.
Response 17: To ensure the accuracy of the data, each sample was tested twice. The relative standard deviation was below 5%, and the data in this paper were the average values of the two tests.
Point 18: Eqs 6 to 11: It is a bit difficult to follow the derivation of the empirical expression. I suggest the authors provide more explanations for each of the steps. For instance, it’s impossible to understand why Eq. 11 is multiplied by 86400 when going from Eq. 10 to Eq. 11
Response 18: Thanks for your suggestion. On the right side of the equation, the unit of is 1/s, but the unit of K is 1/d on the left side. In order to make the dimension of both sides are equal, we have to transfer the unit measured by second to the unit measured by day. And this is why we need to multiplied by 86400 on the right side of the equation. With respect to the transformation of the equation, we have added more explanations for each of the steps.
Point 19: Line 181: “This urban river has conditions similar to the sampled river”. This means that the empirical formula has only been tested on very similar flow, temperature and water quality conditions. How generalizable is the equation to other regions? Is it still valid under different environmental conditions? I understand that the authors cannot possibly contemplate all the possible scenarios in the lab, but this needs to be properly discussed in the “Discussion Section” so that the reader can understand the strengths and weaknesses of the method and formulae proposed.
Response 19: In this paper, we selected the Chishui River as a validation. The Chishui River exhibits the typical characteristics of urban rivers, and is less affected by pollution. The value of predicted displays consistent with the value of observed, which shows that the formula has good applicability. Obviously, urban rivers are affected by many environmental factors, which might cause prediction errors. At present, the formula is mainly applied to the typical urban rivers. In the research, we also discussed many environmental factors may lead to inaccurate predictions, and hope the method could provide a better reference for predicting the water quality in the future.
Point 20: Line 185: The river was monitored on three different occasions between Jan and June 2017, and this is the data that was used to test the empirical equation on field conditions. Three measurements are clearly insufficient to make any trustworthy assessment of the method.
Response 20: In the verification part, we selected three different observation results to test the empirical equation, and the three different observation results are in three different months represent the wet season, temperate season and dry season of a year, respectively. According to previous study (Qiu et al. 2014; Feng et al. 2016), the Chishui River is less affected by pollutants, resulting in the variation of water quality in the same period is not dramatic. So we believe the result is representative when verifying the equation.
Qiu, L., & Zhai, H. J. (2014). An ecological compensation mechanism of Chishui River water resources protection and research. In Applied Mechanics and Materials (Vol. 685, pp. 463-467). Trans Tech Publications.
Feng, Q., Han, L., Tan, X., Zhang, Y., Meng, T., Lu, J., & Lv, J. (2016). Bacterial and archaeal diversities in Maotai section of the Chishui River, China. Current microbiology, 73(6), 924-929.
Point 21: Eq. 12: This equation is not a finding of this work. Therefore, it should not be in the section “Results”. Instead, it should in a sub-section of “Materials and Methods” that explains the methodology adopted to test the empirical equation in field conditions.
Response 21: Thank you for your suggestion. We used the field observation method to determine the degradation coefficient, and the equation provided us an appropriate method. If we move the equation to the section of “Materials and Methods”, it would be isolated from the verification part.
Point 22: Line 204: The statement “Analysis suggested… degradations in the river” is vague, unclear and not a finding of this work. Move it to the Discussion section if appropriate.
Response 22: Thanks for your suggestion. We have moved it to the Discussion section.
Point 23: Line 219-223: The authors did not include any analyses on the effect of sedimentation and chemical analysis of the water besides COD. In fact, this is the first time that sedimentation is mentioned in the paper. So, this is not a finding of the paper. It can be used to discuss the results, but the authors need to make a better job at integrating their findings with existing literature on the topic.
Response 23: We are sorry for not measured the content of sediment in each scenario, and the water samples were not saved, so that it could not be measured again. According to the reference (Lin et al. 2018, Feng et al. 2016), the content of sediment in the Jingjiang river and Chishui river are both low, which has a less influence on degradation coefficient. Thus, the the predicted value is consistent with the observed value.
Liu, X., Jiang, J., Yan, Y., Dai, Y., Deng, B., Ding, S., ... & Gan, Z. (2018). Distribution and risk assessment of metals in water, sediments, and wild fish from Jinjiang River in Chengdu, China. Chemosphere, 196, 45-52.
Feng, Q., Han, L., Tan, X., Zhang, Y., Meng, T., Lu, J., & Lv, J. (2016). Bacterial and archaeal diversities in Maotai section of the Chishui River, China. Current microbiology, 73(6), 924-929.
Point 24: Line 223-226: This statement doesn’t make any sense and, in my opinion, is not adequately linked to the results of this paper.
Response 24: Thanks for your comment. We have revised this part.
Point 25: Line 230-231: What do you mean by “the pollutant source… is fixed…”? Please be more precise.
Response 25: This river reach is located in a natural protection area of valuable and rare fish of the Yangtze River. The pollution mainly comes from domestic pollution, which consists of some fixed pollutants. We have revised the sentence in Line 253-254.
Point 26: Line 238-240: “… is discussed” But this is the section where all should be discussed.
Response 26: We are sorry for our improper expression. In this section, we have discussed the factors that may affect the degradation coefficient process. This paper mainly considers the relationship between hydraulic conditions and degradation coefficient, and we analysed the uncertain factors that may affect the hydraulic conditions in this part.
Point 27: Line 240: what do you mean by “this approach may disrupt the degradation process”? please clearly explain why and how.
Response 27: We have discussed the influence in line 275-282. The experiments were conducted in a circular flume with a DC length of 4m and an arc length of 1.4m, which can be different from the curving ratio of the natural river, thus resulting in reduced accuracy of the prediction.
TECHNICAL NOTES AND SUGGESTIONS:
Point 28: Line 17: change “controllable” to “controlled”
Response 28: Thanks for your suggestion. We have changed it in Line 17.
Point 29:: change “levels” by “depths”
Response 29: As you suggested. We have changed it in Line 31.
Point 30: Line 31: unclear “small gradient ratios” means. The ratio of what by what? Clarify
Response 30: We are sorry for out inappropriate expression. We want describe the elevation of river base changed slightly with distance. We have changed “small gradient ratios” to “small longitudinal gradient”.
Point 31: Line 31: what is flow replacement? Clarify.
Response 31: We are sorry for our uncertain expression. We have changed “flow replacement” to “water exchange effect” in Line 32.
Point 32: Line 32: suggest replacing “serious” by “problematic”
Response 32: As you suggested. We have changed “serious” to “problematic”.
Point 33: Line 39: remove “level”
Response 33: As you suggested. We have removed it.
Point 34: Line 98: remove “to keep… stable.” – it is unnecessary
Response 34: As you suggested. We have removed the sentence.
Point 35: Line 39: replace “Amongst these factors” by “In addition to these environmental factors”
Response 35: Thanks for your suggestion. We have mentioned several factors that might influence the COD degradation, and the hydraulic conditions are the key factors, we mainly discussed the relationship between the hydraulic conditions and the COD degradation. So it is suitable in this part.
Point 36: Line 40: replace “highly” by “strongly”
Response 36: As you suggested. We have replaced it.
Point 37: Line 42: “estimations” instead of “estimation” AND “analyses” instead of “analysis” AND “experiments” instead of “experiment”
Response 37: Thanks for your suggestion. We have revised these.
Point 38: Line 44: suggest adding “In turn, indoor experimental methods…”
Response 38: Thanks for your suggestion. We have added the part.
Point 39: Line 45: suggest changing to “… testing, flow conditions and pollution scenarios. Thus, the experimental conditions may be different from…”
Response 39: As you suggested. We have changed the sentence.
Point 40: Line 51: replace “various” with “different”
Response 40: As you suggested. We have changed “various” to “different”.
Point 41: Line 51: change to “, which can be difficult to represent in laboratory conditions”
Response 41: As you suggested. We have changed the sentence.
Point 42: Line 54: change to “The study of pollution degradation and estimation of degradation rates based on…”
Response 42: Thanks for your suggestion. We turned the noun into a verb form according to the structure of the sentence in Line 55-56.
Point 43: Line 58: add “and uncertainty” after “large prediction errors”
Response 43: As you suggested. We have added “and uncertainty” after “large prediction errors”.
Point 44: Line 60: add “flow” before “velocity”
Response 44: As you suggested. We have changed “various” to “different”.
Point 45: Line 71: remove “the” before “flow velocity”
Response 45: As you suggested. We have deleted it.
Point 46: Line 71: replace “neglect” by “neglects”
Response 46: Thanks for your suggestion. We are sorry for our incorrect writing. We have revised it.
Point 47: Line 77: suggest replacing “paper” by “study” AND replacing “sample raw” by “sampled raw”
Response 47: Thanks for your suggestion. We have changed the sentence.
Point 48: Line 80: change to “was investigated”
Response 48: As you suggested. We have changed it.
Point 49: Line 84: add “the” before “Sichuan”
Response 49: Thanks for your suggestion. We have added it.
Point 50: Line 87: change to “simulate natural river conditions”
Response 50: As you suggested. We have changed the sentence.
Point 51: Line 87: Replace “The hydrodynamic force on” by “Flow at” – be direct and concise
Response 51: Thanks for your suggestion. We have revised the expression.
Point 52: Line 89: add “conditions” after “smooth flow”
Response 52: Thanks for your suggestion. We have revised the expression.
Point 53: Line 99: this is a continuation of the previous sentence – remove the paragraph
Response 53: Thanks for your suggestion. We have divided this part to a new sub-section.
Point 54: Line 113: I suggest removing “and analyses of experiment”
Response 54: As you suggested. We have removed it.
Point 55: Line 132: remove “new paragraph”
Response 55: Thanks for your suggestion. We have built a new sub-section to introduce the methods for monitoring and measurement. This paragraph is used to describe the monitoring method, and we have adjusted the paragraph to the new section.
Point 56: Table 4: replace “of” to “at” in the header columns 7 and 8
Response 56: Thanks for your suggestion. We have replaced it.
Point 57: Table 4: change the date formatting to something more standard and intuitive. For instance, replacing “.” by “-“ would already help. The same things in Table 5
Response 57: Thanks for your suggestion. We have changed the data formatting in the paper.
Point 58: Line 197: There is no Eq. 14. Please correct this.
Response 58: We are sorry for the error. We have corrected it.
Point 59: Line 198: add “for temperature” before “by Eq. (5)”.
Response 59: As you suggested. We have added it.
Point 60: Line 201: Replace “According to the results, it can be seen that the” with simply “The”
Response 60: As you suggested. We have simplified the sentence.
Point 61: Line 210: “many influential factors”? maybe replace with “many environmental factors”?
Response 61: Thanks for your suggestion. We have changed it.
Point 62: Line 210: change to “…many environmental factors that can affect the COD degradation coefficients”
Response 62: As you suggested. We have changed the sentence.
Point 63: Line 212: change to “… 1/d depending on the temperature”
Response 63: We mainly focus on the relationship between hydraulic conditions and the COD degradation coefficient, and the obtained degradation coefficient is corrected by the temperature correction formula. So we considered the degradation coefficient depending on the hydraulic conditions.
Point 64: Line 212: change to “calculated COD degradation coefficients ranging…”
Response 64: As you suggested. We have changed the sentence in Line 244.
Point 65: Line 217: need to add a reference to the statement “Different sources… COD degradation coefficients”
Response 65: Thanks for your reminding. We have noticed the sentence and added the reference in line 242.
Point 66: Line 230: replace “will be” with “becomes”
Response 66: As you suggested. We have changed it.
Point 67: Line 230: replace “pollutant” with “pollution”
Response 67: Thanks for your suggestion. We have replaced it.
Point 68: Line 232: replace “much” with “significantly”
Response 68: As you suggested. We have changed “much” to “significantly”.
Point 69: Line 232: concentration of what?
Response 69: We have specified it is the initial COD concentration.
Point 70: Line 234: what pollutants? Vague statement. Please give some examples at least and provide references for the readers to follow up if they desire
Response 70: Thanks for your reminding. We have added some examples in Line 262-263.
Point 71: Line 242: replace the sentence to something like “This study analyzed the impact of different hydraulic conditions on COD degradation rates/coefficients.
Response 71: As you suggested. We have changed the sentence.
Point 72: Line 248: remove “of the COD degradation coefficient”. The readers know this by now.
Response 72: Thanks for your suggestion. We have deleted the redundant description.
Point 73: Line 250: add “determined” after “those”
Response 73: As you suggested. We have added it in Line 285.

Reviewer 3 Report
This paper is about the experimental study of the COD degradation of water from urban rivers. The sampled river water is filled in a recirculating laboratory flume was used for determining the COD variation and its first-order decay coefficient. The results were analysed and a relationship of the COD decay coefficient with the flow characteristics is developed. The relationship is tested with the field measurement of COD degradation coefficient of another river independently.
The work is interesting and original in the sense that it uses a laboratory circulating flume to resemble the actual flow conditions instead of just measuring the COD in sampled water in a beaker test. The paper is generally well written and the experimental method clearly stated. I would recommend minor revision.
Specific comments:
1. How many samples are obtained at the sampling cross-section? How many repetitions of COD measurement are carried out for each sample?
2. From the result it can be seen that K_COD increases with the flow velocity. It should be increasing with both Froude number and Reynolds number. Could you suggest a reason for the K_COD to be reversely proportional to the Reynolds number in the relationship?
3. Since river flows are mostly turbulent with a high Reynolds number, most of the time we think Reynolds number is not important so what are the reasons for including the Reynolds number here in the K_COD relationship?
4. In Table 3, for Case 1 and 6, since the experiment velocity is zero, why there is a small velocity shown in the column of u/h? Indeed the coefficient c should be the measured K_COD at u = 0.
5. Is there measurement of other water quality parameters such as pH, nutrients, suspended solids to explain the variation in COD or K_COD?
6. It needs some improvement in English in the Abstract and Introduction, as some of the wordings are not quite appropriate: e.g. flow replacement? should it be low flushing? gradient ratios > longitudinal gradient
Author Response
Thank you for your suggestions and comments, which are very helpful for improving our paper. We have revised carefully our manuscript based on the opinions and suggestions.
Point 1: How many samples are obtained at the sampling cross-section? How many repetitions of COD measurement are carried out for each sample?
Response 1: To ensure the accuracy of the data, each sample was tested twice. The relative standard deviation was below 5%, and the data in this paper were the average values of the two tests.
Point 2: From the result it can be seen that K_COD increases with the flow velocity. It should be increasing with both Froude number and Reynolds number. Could you suggest a reason for the K_COD to be reversely proportional to the Reynolds number in the relationship?
Response 2: We conducted a multi-factor method to fit the empirical equation. The fitting formula is not only related to Re, but also related to u, h and Fr. Among the relate parameters, Re and Fr are both related to the flow velocity. When the flow velocity increase, Fr increases faster than Re. Generally, the KCOD increase with the Reynolds number increases.
Point 3: Since river flows are mostly turbulent with a high Reynolds number, most of the time we think Reynolds number is not important so what are the reasons for including the Reynolds number here in the K_COD relationship?
Response 3: Re represents the ratio of the inertia force to the viscous force, which can be an indicator of the flow status. Although urban rivers are turbulence in most situations, the Reynolds number can be used to determine the intensity of turbulence, which can be a guidance to judge the pollutant degradation.
Point 4: In Table 3, for Case 1 and 6, since the experiment velocity is zero, why there is a small velocity shown in the column of u/h? Indeed the coefficient c should be the measured K_COD at u = 0.
Response 4: Accurately there is a tiny velocity of 0.001 m/s for Case 1 and Case 6. In the last manuscript, the hydrodynamic condition was simplified, and the velocity was assumed to be 0. We have corrected the velocity to the accurate value in the new manuscript.
Point 5: Is there measurement of other water quality parameters such as pH, nutrients, suspended solids to explain the variation in COD or K_COD?
Response 5: You have put forward a good suggestion that these indicators are good indicators for reflecting the situation of pollutants. We are so sorry for not measuring the other water quality parameters in the experiments. In the future, we will extend the research to microbial response, and combine with pH, nutrients, suspended solids and flow field distribution.
Point 6: It needs some improvement in English in the Abstract and Introduction, as some of the wordings are not quite appropriate: e.g. flow replacement? should it be low flushing? gradient ratios > longitudinal gradient
Response 6: Thanks for your suggestion. We have revised the improper expression. In order to improve the logic and language accuracy of the paper, we have sent the article to an editing company to make the expression closer to the native speaker. We hope the article could achieve the level suitable for publication.
